# Specific Features of Mitochondrial Dysfunction under Conditions of Ferroptosis Induced by *t*-Butylhydroperoxide and Iron: Protective Role of the Inhibitors of Lipid Peroxidation and Mitochondrial Permeability Transition Pore Opening

**DOI:** 10.3390/membranes13040372

**Published:** 2023-03-24

**Authors:** Tatiana Fedotcheva, Nikolai Shimanovsky, Nadezhda Fedotcheva

**Affiliations:** 1Science Research Laboratory of Molecular Pharmacology, Medical Biological Faculty, Pirogov Russian National Research Medical University, Ministry of Health of the Russian Federation, Ostrovityanova St. 1, Moscow 117997, Russia; 2Institute of Theoretical and Experimental Biophysics, Russian Academy of Sciences, Institutskaya str., 3, Pushchino 142290, Russia

**Keywords:** ferroptosis, tert-butyl hydroperoxide, mitochondrial permeability transition pore, swelling, bromoenol lactone, butylhydroxytoluene, quinidine

## Abstract

Recent studies have indicated the critical importance of mitochondria in the induction and progression of ferroptosis. There is evidence indicating that tert-butyl hydroperoxide (TBH), a lipid-soluble organic peroxide, is capable of inducing ferroptosis-type cell death. We investigated the effect of TBH on the induction of nonspecific membrane permeability measured by mitochondrial swelling and on oxidative phosphorylation and NADH oxidation assessed by NADH fluo rescence. TBH and iron, as well as their combinations, induced, with a respective decrease in the lag phase, the swelling of mitochondria, inhibited oxidative phosphorylation and stimulated NADH oxidation. The lipid radical scavenger butylhydroxytoluene (BHT), the inhibitor of mitochondrial phospholipase iPLA2γ bromoenol lactone (BEL), and the inhibitor of the mitochondrial permeability transition pore (MPTP) opening cyclosporine A (CsA) were equally effective in protecting these mitochondrial functions. The radical-trapping antioxidant ferrostatin-1, a known indicator of ferroptotic alteration, restricted the swelling but was less effective than BHT. ADP and oligomycin significantly decelerated iron- and TBH-induced swelling, confirming the involvement of MPTP opening in mitochondrial dysfunction. Thus, our data showed the participation of phospholipase activation, lipid peroxidation, and the MPTP opening in the mitochondria-dependent ferroptosis. Presumably, their involvement took place at different stages of membrane damage initiated by ferroptotic stimuli.

## 1. Introduction

Ferroptosis is a type of cell death that has been actively studied in recent years. The specific signs of ferroptosis are the appearance of lipid radicals and free irons, each provoking damage to cellular and mitochondrial membranes that underly this form of cell death. Ferroptosis is inherent in various pathologies, such as inflammation, bacterial infections and sepsis, neurodegenerative diseases, ischemia-reperfusion injury, and others [1]. The induction of ferroptosis is mediated by many factors and signaling pathways, including intracellular iron overload, free radical production, lipid peroxidation [2], the suppression of glutathione biosynthesis and glutathione peroxidase activity [3], the activation of lipoxygenase pathways [4], as well as the disturbance of expression and activity of multiple redox-active enzymes that produce or detoxify free radicals and lipid oxidation products [5]. It has been proposed that the main inducers of ferroptosis in vivo are cystine starvation, glutathione depletion, and lipid peroxidation, as well as glutathione-dependent peroxidase inhibition [6].

Although lipid peroxidation leads to the damage of both cellular and mitochondrial membranes and their rupture, recent studies indicate the critical importance of mitochondria in the induction and development of ferroptosis due to their dominant role in the production of reactive oxygen species and, as a result, of lipid radicals in their membranes [7,8]. In addition, mitochondria are at the center of metabolism and consumption of iron, which is necessary for the functioning of a number of redox-dependent enzymes and the respiratory chain. As in the cytosol, glutathione peroxidase inactivation and the depletion of glutathione or its amino acid precursors can promote mitochondrial-dependent ferroptosis [7]. It has recently been shown that, in addition to these factors, various mitochondrial metabolic pathways are involved in the induction and development of ferroptosis, in particular, glutaminolysis, the TCA cycle, and biosynthesis of fatty acid precursors [9,10]. Additionally, a novel ferroptosis activation pathway associated with mitofusin-dependent mitochondrial fusion has recently been identified. It was shown that the classic ferroptosis inducer erastin triggers mitochondrial fusion, which leads to subsequent reactive oxygen species production and lipid peroxidation [11]. As is known, the dynamic regulation of the mitochondrial network by mitofusins modulates energy production, cell survival, and many intracellular signaling events [12]. In addition to mitochondrial fusion, mitofusins are involved in tethering the mitochondria to the endoplasmic reticulum and releasing calcium [13]. It is assumed that the regulation of ferroptosis involves the interaction between subcellular organelles, such as mitochondria, endoplasmic reticulum, lysosomes, lipid droplets, and peroxisomes [11]. Therefore, mitochondria are now considered promising targets in the prevention of cell death through blocking ferroptosis in some pathologies, such as inflammation [14], or conversely, restraining the growth and progression of cancer cells through the induction of ferroptotic cell death [2].

In addition to those ferroptosis inducers that affect the synthesis and transport of glutathione as well as glutathione peroxidase activity, certain drugs, and damaging factors can also induce mitochondrial-dependent ferroptosis. Examples of this include doxorubicin, which knocks out iron from Fe-S clusters [15,16,17], and acidosis, which promotes the release of iron from lysosomes and their weakly bound forms [18]. Respectively, iron chelators such as deferoxamine and lipophilic radical-trapping antioxidants such as ferrostatin-1 lower the levels of labile iron and lipid radicals, thereby inhibiting ferroptosis [4].

Recent studies have shown that *tert*-butyl hydroperoxide (TBH), a lipid-soluble organic peroxide, is capable of inducing ferroptosis-type cell death. So, it was found that mitochondrial dysfunction was involved in the TBH-induced death of PC12 neural cells, and both these events were prevented by ferrostatin-1 and deferoxamine [19]. It is noteworthy that the occurrence of the protective effect of ferrostatin-1 is currently considered to be an indicator of ferroptosis since this antioxidant is unable to inhibit other common forms of cell death, including apoptosis [4,6]. Additionally, TBH-induced cell death in murine fibroblasts and human keratinocytes cell lines were identified as ferroptosis due to the ferrostatin-1 protection [20]. TBH treatment also decreased the viability of macrophages in a dose- and time-dependent manner via a mitochondria-mediated signaling pathway [21]. It is believed that the initiation of ferroptosis requires only a few radicals to start the autocatalytic chain reaction [6]. Lipophilic organic peroxide TBH as an analog of lipid peroxide, is the most frequently used agent to activate lipid peroxidation. Lipid peroxidation products formed by TBH, e.g., malondialdehyde or 4-hydroxynonenal, are pathophysiologically relevant [20]. The initiation and propagation of lipid peroxidation include the formation of lipid peroxyl radicals, lipid peroxides, and lipid alkoxyl radicals [4]. Initiating radicals can be lipid alkoxyl radicals or hydroxyl radicals derived from the one-electron reduction of hydroperoxide in a Fenton reaction, which is ferrous ion-dependent. In addition to iron, the reductants can be other endogenous electron donors [6]. 

Earlier, in studies on isolated mitochondria, we have shown that TBH causes a concentration-dependent decrease in membrane potential, which was enhanced by the addition of low concentrations of iron [22]. In addition, we have shown that loading with iron ions and acidification lead to the activation of the nonspecific permeability of mitochondrial membranes [23]. We have also observed similar effects when studying the influence of a complex of doxorubicin with iron on mitochondria [24]. In the present work, we investigated the effect of iron, TBH, and their combinations on the induction of nonspecific membrane permeability measured by the mitochondrial swelling assay. We compared the influence of the inhibitors of lipid peroxidation, including ferrostatin-1, and the inhibitors of mitochondrial permeability transition pore opening to estimate the involvement of these two processes in the effect of TBH and iron. We also examined their effect on oxidative phosphorylation and NADH oxidation and evaluated the protective effect of inhibitors on these parameters.

## 2. Materials and Methods

### 2.1. Reagents and Chemicals

All reagents were from the Sigma–Aldrich Corporation (St. Louis, MO, USA).

### 2.2. Preparation of Rat Liver Mitochondria

Mitochondria were isolated from adult Wistar male rats. The study was conducted in accordance with the ethical principles formulated in the Helsinki Declaration on the care and use of laboratory animals. Manipulations were carried out by the certified staff of the Animal Department of the Institute of Theoretical and Experimental Biophysics (Russian Academy of Sciences and approved by the Commission on Biomedical Ethics of ITEB RAS (N2/2022, 3 May 2022). Mitochondria from the liver was isolated using the standard method. The liver was rapidly removed and homogenized in an ice-cold isolation buffer containing 300 mM sucrose, 1 mM EGTA, and 10 mM HEPES–Tris (pH 7.4). The homogenate was centrifuged at 600× *g* for 7 min at 4 °C, and the supernatant fraction was then centrifuged at 9000× *g* for 10 min to obtain the mitochondria. Mitochondria were washed twice in the above medium without EGTA. The final mitochondrial pellet was suspended in the washing medium to yield 60 mg protein/mL and was kept on ice for analysis. 

### 2.3. Estimation of Swelling of Mitochondria

The swelling of mitochondria was measured at a wavelength of 540 nm using an Ocean Optics USB4000 spectrophotometer permitting general UV and visible measurements in a wide wavelength range, from 200 to 850 nm, and communicating with a computer via SpectraSuite Software (Ocean Optic, Dunedin, FL, USA). The swelling was assessed by measuring the changes in optical density during incubation. Mitochondria at a concentration of mitochondrial protein at 0.3–0.4 mg/mL were incubated in the buffer containing 125 mM KCl, 15 mM HEPES, 1.5 mM phosphate, and 5 mM succinate, as described earlier [25].

### 2.4. Estimation of Oxidation of Succinate and NAD-Dependent Substrates by the Methyl Thiazolyl Tetrazolium (MTT) Assay

An incubation medium (2 mL) containing 125 mM KCl, 20 mM HEPES, pH 7.4, 150 µM MTT, and the oxidation substrate were mixed with the mitochondria (0.5 mg protein per mL) and incubated for 5 min as described earlier [26]. The samples were placed simultaneously in a series of spectrophotometric cuvettes. The reaction of acceptor reduction was initiated by the addition of mitochondria. Subsequent to incubation, mitochondria were lysed by Triton X-100 (10 µL of 10% solution, and optical density was recorded at 580 nm with a USB4000 spectrophotometer (Ocean Optic, Dunedin, FL, USA).

### 2.5. Determination of the Redox State of Pyridine Nucleotides and Oxidative Phosphorylation in Mitochondria

The redox state of pyridine nucleotides and oxidative phosphorylation in mitochondria in suspension was determined by recording the fluorescence of pyridine nucleotides (excitation at 340 nm, emission at 460 nm) on a Hitachi-F700 fluorimeter (Japan), as described earlier [26]. Mitochondria (0.6 mg protein/mL) was added to the medium containing 125 mM KCl, 1.5 mM KH_2_PO_4,_ and 15 mM HEPES–Tris (pH 7.25), and glutamate with malate were added as the substrates of oxidation. ADP (100 µM) was added to evaluate oxidative phosphorylation. The complete oxidation of pyridine nucleotides was induced by adding the uncoupler FCCP (0.5 µM).

### 2.6. Statistical Analysis

The data given represent the means ± standard error of means (SEM) from five to seven experiments or are the typical traces of three to five identical experiments with the use of different mitochondrial preparations. The statistical significance was estimated using Student’s *t*-test with *p* < 0.05 as the criterion of significance.

## 3. Results

### 3.1. Influence of TBH and Ferrous Ions on the Swelling of Mitochondria

Previously, we tested the influence of iron and TBH separately on the induction of mitochondrial swelling. As shown in Figure 1a, iron (50 µM FeCl_2_) induced the swelling of mitochondria only after a long incubation, for about 600 s. A significantly shorter incubation time was required to induce swelling by TBH (100 µM). When the compounds were added sequentially at the same concentrations, the swelling was induced several times faster, within 200 s. A similar fast effect was achieved by varying the concentrations of the compounds, namely, by decreasing the concentration of one with a simultaneous increase in the concentration of the other (Figure 1b, inset). In all the following experiments, we used 50 μM FeCl_2_ and 100 μM TBH to obtain a lag phase that was optimal for studying the action of the inhibitors of swelling, which was estimated by the duration of the lag phase and the swelling rates.

First of all, we tested the effect of ferrostatin as an indicator of ferroptotic changes in the mitochondria. It turned out that ferrostatin delayed the induction of swelling by 1.5 and 2 times as its concentration was increased from 25 to 50 μM (Figure 1c). Ferrostatin also reduced the swelling rate by 20% and 40% at these concentrations. Thus, ferrostatin had an effect on the isolated mitochondria, acting as a lipophilic lipid radical-trapping compound. However, another lipid radical scavenger, butylhydroxytoluene (BHT), was more effective. As shown in Figure 1d, at a concentration of 20 μM, BHT increased the lag phase by almost five times. An even stronger inhibitory effect was observed upon incubation with bromoenol lactone (BEL), an inhibitor of phospholipase, which completely prevented the induction of swelling, at least for more than 600 s of the recording (Figure 1d). Thus, in this group of inhibitors associated with the suppression of lipid peroxidation, BEL was the most effective, followed by BHT and ferrostatin.

In the next experiments, we examined the influence of inhibitors affecting the cyclosporine-sensitive permeability transition pore (MPTP). Figure 1 shows the influence of ADP, CsA, and oligomycin on the swelling induced by ferrous ions and TBH. CsA completely prevented the induction of swelling, while ADP and oligomycin increased the lag phase by 1.5 and 2 times, respectively. Both compounds also reduced the rate of swelling. Moreover, their effect was greatly increased when they were added together (Figure 1f). In this case, the swelling rate decreased by 2.5 times, while the lag phase increased almost threefold. Figure 1g,h also demonstrates the protective effects of the inhibitors in terms of the swelling delay as the most obvious indicator. It can be concluded that the most effective inhibitors were BEL, BHT, and CsA.

### 3.2. Influence of TBH and Ferrous Ions on the Oxidation of Succinate and NAD-Dependent Substrates and Oxidative Phosphorylation

To evaluate the possible contribution of changes in the oxidative activity of mitochondria, in the following experiments, we checked the influence of TBH, ferrous ions, and their combination on the oxidation of succinate and NAD-dependent substrates using MTT as an electron acceptor. Figure 2 shows MTT reduction during the oxidation of succinate (a) and glutamate with malate (b). TBH, only in combination with iron, caused a slight decrease in the reduction of MTT during the oxidation of succinate and did not affect the oxidation of glutamate with malate.

However, they had a strong influence on oxidative phosphorylation, measured by changes in the redox state of pyridine nucleotides in response to ADP supplementation during the oxidation of glutamate with malate. As shown in Figure 3a, TBH significantly increased the phosphorylation time, decreased the level of NAD reduction after ADP phosphorylation, and, when supplemented with ferrous ions, completely inhibited oxidative phosphorylation. This effect was tested for susceptibility to inhibitors of both lipid peroxidation and the MPTP opening.

It turned out that the inhibitors of lipid peroxidation, BEL, and BHT, as well as CsA, as an inhibitor of the MPTP opening, had a protective effect against the inhibition of oxidative phosphorylation caused by TBH and iron (Figure 3b). In the presence of these inhibitors, an almost complete recovery of NADH was observed after ADP phosphorylation, as also evidenced by the typical effect of FCCP: an uncoupler of oxidative phosphorylation. The influence of ferrostatin could not be tested because the compound showed its own high fluorescence in this range.

During the continuous incubation of mitochondria with TBH and iron, a decrease in the NADH level was observed with time (Figure 3c). In these cases, all three inhibitors effectively reversed the drop in NADH. Interestingly, oligomycin also suppressed a fall in NADH, and its effect was enhanced by the presence of ADP (Figure 3d). Compared with the control in the absence of inhibitors, the time of the decrease in the level of NADH until its complete oxidation increased twofold in the presence of oligomycin and threefold in the presence of ADP and oligomycin. To elucidate the contribution of the swelling and MPTP opening and the resulting release and dilution of NADH compared to the decrease in the NADH level, rotenone was added to inhibit the oxidation of NADH in the respiratory chain. Indeed, as shown in Figure 3e, both processes, namely, rotenone-sensitive NADH oxidation and the rotenone-insensitive MPTP opening, were equally responsible for the fall in the NADH level induced by TBH and iron. As shown in Figure 3e, the insert, iron, and TBH-induced mitochondrial swelling during the oxidation of glutamate plus malate produced a shorter lag phase than during the oxidation of succinate. As with succinate, ADP slowed down swelling and increased the lag phase, while CsA almost completely inhibited swelling. Thus, these experiments, on the one hand, confirm the data presented in Section 3.1. and indicate the involvement of NADH oxidation in the other hand. Therefore, the influence of inhibitors extends to both of these processes.

### 3.3. Other Protectors against the Mitochondrial Dysfunction Induced by TBH and Ferrous Ions 

We have previously shown that the alkalization of the medium protects mitochondria against MPTP opening under loading with iron ions [20]. In the next experiment, we tested the effect of alkalization on mitochondrial swelling induced by TBH with iron ions. Under these conditions, increasing the pH level to 7.8 inhibited swelling (Figure 4a) but only partially protected oxidative phosphorylation (Figure 4b). Thus, alkalization showed a less pronounced effect than the above-tested specific inhibitors.

One more inhibitor we have previously tested as a regulator of the Ca^2+^-induced MPTP opening was the membrane-active compound quinidine [22]. As shown in Figure 4c, when swelling was induced by TBH with iron, quinidine increased the pre-swelling lag by only 40%. However, the protective effect of quinidine increased significantly if the swelling was induced by TBH alone. In this case, quinidine increased the lag phase by 1.5 and 2.5 times at concentrations of 100 and 200 µM, respectively, and also significantly increased this more than two times and reduced the swelling rate (Figure 4d).

## 4. Discussion

The results of the experiments showed the participation of lipid peroxidation, phospholipase activation, and the MPTP opening in the disturbances of mitochondrial functions initiated by ferroptotic stimuli, namely, the lipid radical analog TBH and ferrous ions. These compounds induced the swelling of mitochondria, inhibited oxidative phosphorylation, and stimulated NADH oxidation. The protective effect of ferrostatin as an indicator of ferroptosis indicated that these disorders were related to this type of cell death. This is consistent with the data that mitochondria are a target of ferrostatin action, which was manifested in the restoration of the mitochondrial membrane potential and ATP production upon TBH-induced ferroptosis in cell cultures [19,20].

As shown in isolated mitochondria, TBH also markedly inhibited state 3 respiration [27], as well as decreased the membrane potential and calcium retention capacity [28,29]. In addition, TBH activated the MPTP opening since its action was prevented by CsA [22]. In all studies, in both isolated mitochondria and cell cultures, high concentrations of TBH, from 100 to 400 µM, as well as a prolonged incubation, were required to induce ferroptotic alterations. As follows from our data, the effect of TBH was sharply enhanced in the presence of ferrous ions, which allowed one to diminish both their concentration and incubation time. In addition to the known activation of lipid peroxidation [3,4], ferrous ions sensitized mitochondria to the MPTP opening [30]. Thus, the mitochondrial dysfunction induced by iron and TBH could serve as an adequate model for testing the possible inhibitors of ferroptosis. The measurement of mitochondrial swelling as a criterion for assessing an increase in the permeability of mitochondrial membranes made it possible to evaluate the contribution of the MPTP opening to the membrane damage as the main sign of ferroptosis.

Among the inhibitors, the most effective was the inhibitor of calcium-independent phospholipase BEL, the lipid radical scavenger BHT, and the inhibitor of MPTP opening CsA. Ferrostatin, which, similar to BHT, belongs to lipid radical-trapping antioxidants, acted much weaker. Most interesting is the result showing that all three inhibitors, BEL, BHT, and CsA, were equally effective in protecting mitochondria from undergoing swelling induction, the inhibition of oxidative phosphorylation, and NADH oxidation. Apparently, all three factors, including lipid hydrolysis, the formation of lipid radicals, and the opening of MPTP, were involved in the realization of these disorders. Moreover, their inhibitors could replace each other in the effective protection of these mitochondrial functions.

In this regard, it is important to note that BEL is a selective inhibitor of phospholipase iPLA2γ, which is localized in the mitochondria [31,32]. Phospholipase iPLA2γ was identified as the membrane potential-sensitive phospholipase in liver mitochondria and was activated under conditions of de-energization [31,33]. In these studies, BEL inhibited phospholipase iPLA2γ activation. BEL also blocked the activation of mitoKATP, preventing the release of oxidized phospholipids from the bilayer [34]. As follows from our data, BEL is one of the most effective protectors against ferroptotic disorders in the mitochondria, which indicates the involvement of phospholipase iPLA2γ in mitochondria-dependent ferroptosis, possibly at the initial stage associated with the activation of membrane lipid hydrolysis. If this is the case, then BEL suppresses all subsequent stages of lipid peroxidation in the mitochondria. In turn, BHT, as a radical-trapping antioxidant, blocks the propagation of lipid peroxidation by converting peroxyl radicals to non-radical products, while CsA inhibits the opening of MPTP activated by these membrane processes.

Experiments assessing the redox state of NADH also show the involvement of these processes in the inhibition of oxidative phosphorylation and the activation of NADH oxidation since these effects were also eliminated by BEL, BHT, and CsA, each exhibiting a different degree of efficiency. The inhibition of oxidative phosphorylation and the activation of NADH oxidation occurred only upon severe ferroptosis induced by TBH together with iron and preceded the MPTP opening. Partial elimination of the fall in the NADH level in the presence of rotenone evidenced the contribution of NADH oxidation itself to the overall NADH decline. The portion of the fall in the NADH level that is not inhibited by rotenone is most likely associated with the opening of MPTP, followed by the release of NADH from the matrix and its dilution in the medium. Therefore, it can be assumed that the ability to provoke NADH oxidation is among the features inherent in ferroptosis.

The inhibitory effect of CsA is mediated by cyclophilin D: a regulator of the MPTP opening. It can be assumed that the formation of MPTP was accomplished by several participants, including cyclophilin D, adenine nucleotide translocase, and ATP synthase [35,36]. As follows from our data, ADP and oligomycin significantly decelerated iron- and TBH-induced swelling. While the effect of ADP as an inhibitor of the MPTP opening had been well established, the data on the influence of oligomycin were contradictory. Thus, it is considered that oligomycin does not affect the MPTP opening in native membranes [36]. However, there is also evidence that oligomycin completely blocks the egachannel formed by reconstructing mitochondrial ATP synthase into liposomes [37]. It can be assumed that, under the conditions of ferroptosis, which were imitated in our experiments, membrane disturbances led to an increase in the availability of the oligomycin-sensitive site. Moreover, the effect of oligomycin under these conditions exceeded the effect of ADP, which was much weaker than usually observed in intact mitochondria. This result can be explained by conformational changes to the proteins under the influence of lipid peroxidation.

As for other protective factors, namely the influence of alkalization and membrane-active substances, their effects were weaker compared to BEL, BHT, and CsA. Of interest is the influence of quinidine in the context of its widespread use as an antiarrhythmic drug. According to our data, quinidine can serve as an active protector against moderate ferroptosis, which may be due to its amphiphilic properties. Additionally, the protective effect of quinidine on lipid peroxidation was noted earlier [38].

Thus, the study revealed new protectors against mitochondria-dependent ferroptosis. According to the degree of protection, they could be arranged in the following order: Bromoenol Lactone ≥ Butylhydroxytoluene ≥ CsA ˃ Ferrostatin ˃ Oligomycin ˃ ADP. Of greatest interest in the context of the participation of MPTP is the analysis of the succession of stages of membrane modifications leading to the MPTP opening during ferroptosis progression, which requires further research.

## Figures and Tables

**Figure 1 membranes-13-00372-f001:**
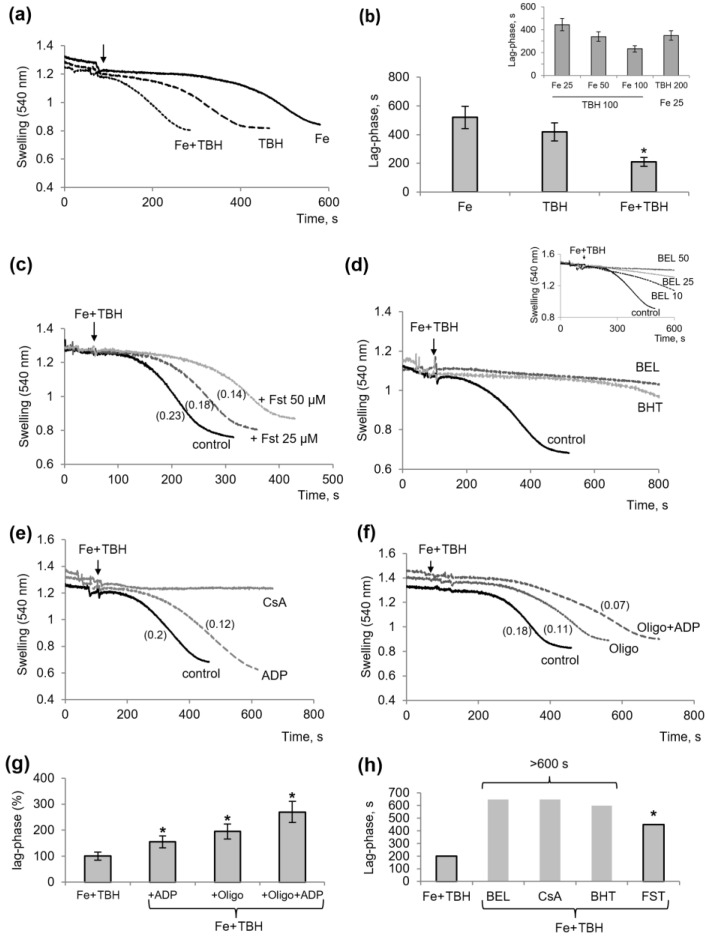
Induction of the mitochondrial swelling by TBH and ferrous ions. Influence of inhibitors of the lipid peroxidation and MPTP opening. Influence of TBH (100 μM), FeCl_2_ (50 μM) on the swelling induction (**a**) and the lag phase at their different concentrations (**b** and insert); the protective influence of ferrostatin (FST) (**c**), BEL (50 μM), BHT (20 μM), and BEL at different concentrations (**d**, insert), ADP and CsA I, and oligomycin (1.5 μM) (**e**,**f**) on the induced swelling. Influence of inhibitors on the pre-swelling time (lag-phase) (**g**,**h**). The swelling rate (Δ/min) is indicated in parentheses. An asterisk (*) indicates values that differ significantly from the control values (*p* < 0.05).

**Figure 2 membranes-13-00372-f002:**
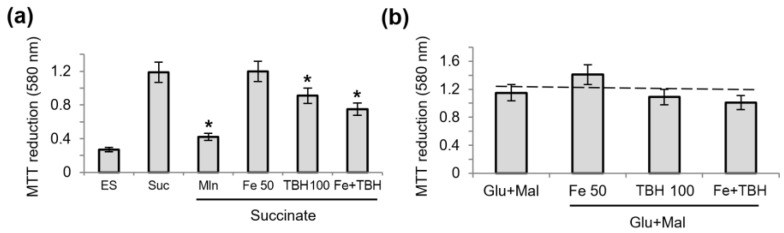
The influence of TBH and ferrous ions on the oxidation of succinate and NAD-dependent substrates. MTT reduction measured by an increase in optical density after the incubation of mitochondria without a substrate (ES) and in the presence of succinate (Suc), malonate (Mln), TBH, and ferrous ions (**a**); the influence of TBH and ferrous ions on MTT reduction during oxidation of glutamate with malate (**b**). Asterisk (*) indicates values that differ significantly from the control values (*p* < 0.05).

**Figure 3 membranes-13-00372-f003:**
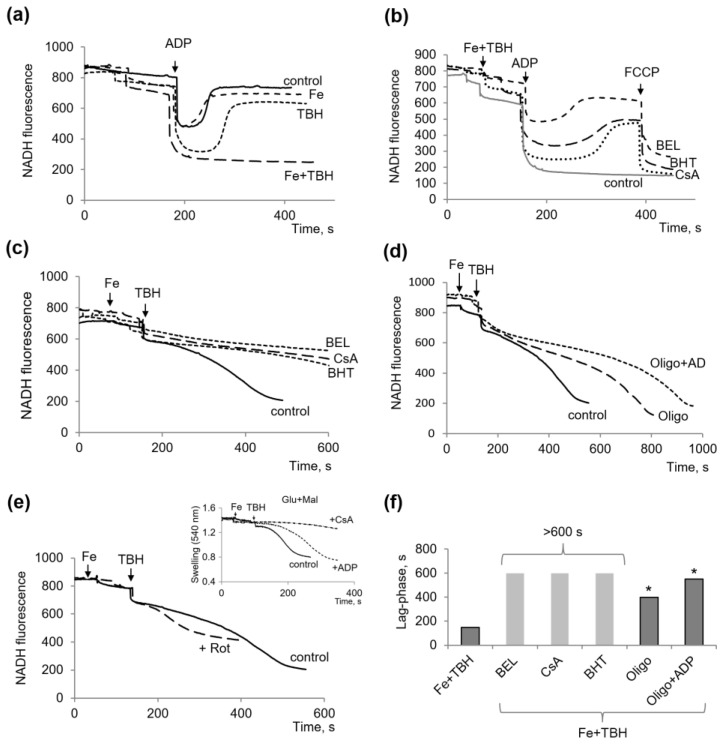
Influence of TBH and ferrous ions on the oxidative phosphorylation measured by NADH fluorescence during oxidation glutamate with malate. Influence of ferrous ions, TBH, and their together on ADP-stimulated changes in the NADH redox state (**a**); the influence of BEL (50 µM), BHT (20 µM), and CsA (1 µM) on oxidative phosphorylation (**b**) and NADH oxidation (**c**) in the presence of TBH with ferrous ions; the influence of oligomycin (1.5 µM), its combination with ADP (100 µM) (**d**) and rotenone (1 µM) (**e**) on NADH oxidation induced by TBH and ferrous ions; the influence of ferrous ions and TBH on the swelling induction during glutamate with malate oxidation (**e**, insert); comparison of the protective effects of inhibitors on the duration of the lag phase preceding the fall in NADH induced by TBH and ferrous ions (**f**). Asterisk (*) indicates values that differ significantly from the control values (*p* < 0.05).

**Figure 4 membranes-13-00372-f004:**
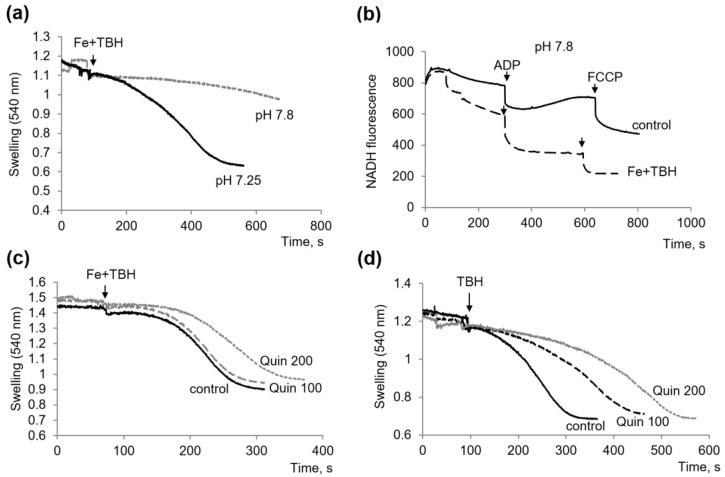
Influence of alkalization and quinidine on mitochondrial damage induced by TBH and ferrous ions. Influence of alkalization (pH 7.8) on the swelling (**a**) and oxidative phosphorylation (**b**) in the presence of TBH and ferrous ions; Influence of quinidine (Quin) at the indicated concentration on the swelling induced by TBH and ferrous ions (**c**) and TBH alone (**d**).

## Data Availability

The datasets generated during the current study are available from the corresponding author upon reasonable request.

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
