# Peer review of "Specific Features of Mitochondrial Dysfunction under Conditions of Ferroptosis Induced by t-Butylhydroperoxide and Iron: Protective Role of the Inhibitors of Lipid Peroxidation and Mitochondrial Permeability Transition Pore Opening"

_membranes, 2023, doi:10.3390/membranes13040372_

Round 1

Reviewer 1 Report

I really like this manuscript. Minor adjustment can improve manuscript:

- In the introduction authors need to add more sentence related also mito-swelling and mitofusions role related to Calcium beside to Ferro. (Ballard A. JBC 2020). They also need to add more details related to mitochondria membrane and Its role in mitochondria defects (Franco A. Life 2022). 

- Material and Methods  paragraph 2.3 please add just two sentence about this technique 

- Fig 1 :panel b and d subscribed graph need to be more bigger (size)

- Fig 2 : Y axes need to change from 0, 0.4,0.8 eccc

Author Response

We thank the reviewer for the useful remarks to our work. We have made appropriate corrections and additions to the manuscript.

- In the introduction authors need to add more sentence related also mito-swelling and mitofusions role related to Calcium beside to Ferro. (Ballard A. JBC 2020). They also need to add more details related to mitochondria membrane and Its role in mitochondria defects (Franco A. Life 2022). 

In the Introduction section, we have added the following paragraph related to the mitofusion role in ferroptosis:

Also, a novel ferroptosis activation pathway associated with mitofusin-dependent mitochondrial fusion has recently been identified. It was shown that the classic ferroptosis inducer erastin triggers mitochondrial fusion, which leads to subsequent reactive oxygen species production and lipid peroxidation [11]. As is known, dynamic regulation of the mitochondrial network by mitofusins modulates energy production, cell survival, and many intracellular signaling events [12]. Besides mitochondrial fusion, mitofusins are involved in tethering mitochondria to the endoplasmic retic-ulum and releasing calcium [13]. It is assumed that the regulation of ferroptosis involves the interaction between subcellular organelles, such as mitochondria, endoplasmic reticulum, lysosomes, lipid droplets, and peroxisomes [11].

Respectively, we have added the following references:

  1. Li, C.; Liu, J.; Hou, W.; Kang, R.; Tang, D. STING1 Promotes Ferroptosis Through MFN1/2-Dependent Mitochondrial Fusion. Front Cell Dev Biol. 2021 Jun 14;9:698679. doi: 10.3389/fcell.2021.698679. eCollection 2021.
  2. Ballard, A.; Zeng, R.; Zarei, A.; Shao, C.; Cox, L.; Yan, H.; Franco, A.; Dorn, G.W.2nd; Faccio, R.; Veis, D.J. The tethering function of mitofusin2 controls osteoclast differentiation by modulating the Ca2+-NFATc1 axis. J Biol Chem. 2020 May 8;295(19):6629-6640. doi: 10.1074/jbc.RA119.012023.
  3. Franco, A.; Walton, C.E.; Dang, X. Mitochondria Clumping vs. Mitochondria Fusion in CMT2A Diseases. Life (Basel). 2022 Dec 15;12(12):2110. doi: 10.3390/life12122110.

- Material and Methods  paragraph 2.3 please add just two sentence about this technique 

In the Material and Methods section we have added we have added some technical characteristics in the paragraph 2.3:

The swelling of mitochondria was measured at a wavelength of 540 nm using an USB4000 spectrophotometer permitting general UV and visible measurements in a wide wavelength range, from 200 to 850 nm, and communicating with a computer via SpectraSuite Software (Ocean Optic, Dunedin, FL, USA)

- Fig 1 :panel b and d subscribed graph need to be more bigger (size)

- Fig 2 : Y axes need to change from 0, 0.4,0.8 eccc

We have also corrected Figures 1 and 2 according to comments.

Reviewer 2 Report

The authors in the manuscript „ Specific Features of Mitochondrial Dysfunction Under Conditions of Ferroptosis Induced by t-Butylhydroperoxide and Iron. Protective Role of the Inhibitors of Lipid Peroxidation and Mitochondrial Permeability Transition Pore Opening“ investigated the effect of iron, TBH, and their combinations on the induction of non-specific membrane permeability measured by the mitochondrial swelling assay. Authors compared the influence of the inhibitors of lipid peroxidation to estimate the involvement of these two processes in the effect of TBH and iron. Authors also examined their effect on oxidative phosphorylation and NADH oxidation and evaluated the protective effect of inhibitors on these parameters.

The topic is interesting, but after careful revision of the manuscript I noted several points which the authors should address when if invited to prepare a revision:

1. Authors analysed the effect of TBH and ferrous ions on the swelling of mitochondria when succinate was used as substrate. Can authors add additional information about the effect of these compounds on swelling of mitochondria when glutamate and malate are used as respiratory substrates. 

2. Authors showed that Fe+TBH decreased MTT reduction when succinate was used Fig. 2A), but there is no information about the effect of these compounds on oxidative phosphorylation. Can authors add additional information which could better explain the mechanism oh TBH and Fe action

Author Response

We thank the reviewer for the useful remarks to our work. We have made appropriate corrections and additions to the manuscript.

  • Authors analysed the effect of TBH and ferrous ions on the swelling of mitochondria when succinate was used as substrate. Can authors add additional information about the effect of these compounds on swelling of mitochondria when glutamate and malate are used as respiratory substrates. 

We have added to the Results section a figure (inset in Fig. 3) showing the induction of swelling by iron and TBH when glutamate with malate are used as oxidation substrates. Also, we made a corresponding description in the text:

As shown in Figure 3e, insert, iron and TBH induced mitochondrial swelling during oxidation of glutamate plus malate, with a shorter lag phase than during oxidation of succinate. As with succinate, ADP slowed down swelling and increased the lag phase, while CsA almost completely inhibited swelling.

  • Authors showed that Fe+TBH decreased MTT reduction when succinate was used Fig. 2A), but there is no information about the effect of these compounds on oxidative phosphorylation.
  • Can authors add additional information which could better explain the mechanism oh TBH and Fe action

Our experience shows that it is practically impossible to measure the effect of iron and TBH on oxidative phosphorylation by the polarographic method (in the case of succinate use) due to the parallel consumption of oxygen in iron redox reactions. Therefore, the measurement of NADH fluorescence is the most acceptable method. The known influence of TBH on oxidative phosphorylation and energy-dependent processes is mentioned in the Discussion with references 27-29.

  • Can authors add additional information which could better explain the mechanism oh TBH and Fe action

The activation of lipid peroxidation as well as the experimental use of TBH and iron to activate the process are very well described in the cited articles, ref. 4,6,20,21 and others. We made a short addition on this topic to the Introduction section:

It is believed that initiation of ferroptosis requires only a few radicals to start the autocatalytic chain reaction [6]. The lipophilic organic peroxide TBH as analog of lipid peroxide is the most frequently used agent to activate lipid peroxidation. Lipid peroxi-dation products formed by TBH, e.g., malondialdehyde or 4-hydroxynonenal, are pathophysiologically relevant [20]. Initiation and propagation of lipid peroxidation include the formation of lipid peroxyl radicals, lipid peroxides, and lipid alkoxyl radicals [4]. Initiating radicals can be lipid alkoxyl radicals or hydroxyl radicals derived from one-electron reduction of the hydroperoxide in a Fenton reaction, which is ferrous ion-dependent. Besides iron, the reductants can be other endogenous electron donors [6].

Round 2

Reviewer 2 Report

Accept